# Performance Analysis of Embedded Multilayer Perceptron Artificial Neural Networks on Smart Cyber-Physical Systems for IoT Environments

**DOI:** 10.3390/s23156935

**Published:** 2023-08-04

**Authors:** Mayra A. Torres-Hernández, Miguel H. Escobedo-Barajas, Héctor A. Guerrero-Osuna, Teodoro Ibarra-Pérez, Luis O. Solís-Sánchez, Ma del R. Martínez-Blanco

**Affiliations:** 1Instituto Politécnico Nacional, Unidad Profesional Interdisciplinaria de Ingeniería Campus Zacatecas, Calle Circuito Cerro del Gato No. 202, Col. Ciudad Administrativa, Zacatecas 98160, Mexico; mtorresh@ipn.mx (M.A.T.-H.); tibarrap@ipn.mx (T.I.-P.); 2Programa de Doctorado en Ingeniería y Tecnología Aplicada, Universidad Autónoma de Zacatecas, Av. Ramón López Velarde No. 801, Col. Centro, Zacatecas 98000, Mexicohectorguerreroo@uaz.edu.mx (H.A.G.-O.); lsolis@uaz.edu.mx (L.O.S.-S.); 3Laboratorio de Inteligencia Artificial Avanzada (LIAA), Universidad Autónoma de Zacatecas, Av. Ramón López Velarde No. 801,Col. Centro, Zacatecas 98000, Mexico

**Keywords:** Industry 4.0, artificial neural networks, multilayer perceptron, smart cyber-physical systems, cloud computing, fog computing

## Abstract

At present, modern society is experiencing a significant transformation. Thanks to the digitization of society and manufacturing, mainly because of a combination of technologies, such as the Internet of Things, cloud computing, machine learning, smart cyber-physical systems, etc., which are making the smart factory and Industry 4.0 a reality. Currently, most of the intelligence of smart cyber-physical systems is implemented in software. For this reason, in this work, we focused on the artificial intelligence software design of this technology, one of the most complex and critical. This research aimed to study and compare the performance of a multilayer perceptron artificial neural network designed for solving the problem of character recognition in three implementation technologies: personal computers, cloud computing environments, and smart cyber-physical systems. After training and testing the multilayer perceptron, training time and accuracy tests showed each technology has particular characteristics and performance. Nevertheless, the three technologies have a similar performance of 97% accuracy, despite a difference in the training time. The results show that the artificial intelligence embedded in fog technology is a promising alternative for developing smart cyber-physical systems.

## 1. Introduction

### 1.1. Industry 4.0 (I4.0)

At present, modern society is experiencing a significant transformation concerning the production of products thanks to the digitization of society and manufacturing [1]. This transition is being called Industry 4.0 (I4.0), which is integrated by new, modern, smart, and disruptive technologies. It refers to the intelligent networking of machines and processes with the help of new information and communication technologies (NTICs) such as the Internet of Things (IoT), cloud computing (CC), fog computing (FC), Artificial Intelligence (AI), among others [2,3,4,5], that have increased the speed and breadth of knowledge within the modern economy and society of knowledge.

From the first industrial revolution, distinguished by mechanization through water and steam power [6], to the mass production and assembly lines using electricity in the second [7], I4.0 adopted computers and automation, which began in the third industrial revolution [8], and enhanced it with smart and autonomous systems, which work in IoT environments, fed by digital data analyzed by machine learning (ML) technology [9].

When computers were introduced in the third industrial revolution, it was disruptive compared to the technology used during the second one [10]. Thanks to the addition of this entirely new technology, at present, computers are connected, communicate with one another, and are capable of making decisions without human involvement because of the NTICs of I4.0 [11,12,13], where cyber-physical systems (CPS), IoT, AI, and ML stand out.

A combination of CPS, IoT, the Internet of Systems (IoS), and ML are making I4.0 possible and the smart factory a reality. With the support of smart machines that access more data to get smarter themselves, the actual factories will be less wasteful and more efficient and productive [14]. Indeed, the true power of I4.0 is provided by the digital connection between network CPS and one another to create and share information.

### 1.2. Cyber-Physical Systems

CPS are systems that integrate constituents from the cyber and physical domains [15,16] for monitoring and controlling the physical processes through a network of actuators and sensors and could be implemented on many scales between the nano-world and large-scale systems [17]. A CP system is a computer system controlled and/or monitored by computer-based algorithms, which integrates networking, computation, and physical processes where embedded networks and computers control and monitor the physical processes, with feedback loops where physical processes affect computations reciprocally [18]. CPS form the basis of smart machines of I4.0 mainly because they use modern control systems and have and dispose of an internet address to connect and be addressed via IoT environments [19].

As previously mentioned, there is great potential for CPS in modern society. Economy and major research and investments are being made worldwide to develop this technology [20] based on older technologies, such as embedded systems, computers, networks, and embedded software.

CPS hold the potential to reshape our world with more precise, responsive, efficient, and reliable systems, enabling a revolution of smart devices and systems ranging from smart cars, medical and household appliances, etc., and passing to smart grids towards smart cities [21].

### 1.3. Smart Cyber-Physical Systems

In computing applications, smart cyber-physical systems (SCPS) are the next generation, in which communication, computation, process control, and AI technologies are integrated in a transparent and novel way, developing intelligent autonomous systems [22,23]. The term smart indicates highly cooperative behavior, self-awareness, self-adaptation, and self-optimization [24].

In this sense, the SCPS are complex due to the combination and composition of the components and physical elements that integrate them, and where the main drawback for becoming smart is the enabling of different AI techniques and technologies under a variety of IoT constraints [25]. Nowadays, new developments are allowing the emergence of new SCPS where the continuously generated data is utilized to build AI models used to perform specialized tasks within the systems, such as image, character, voice recognition, and more applications [26,27]. Therefore, SCPS must be studied as a whole, which sets this emerging discipline apart from the older technologies on which it is based.

SCPS have evolved beyond what was identified by the traditional definitions of CPS. The SCPS describes the upcoming generation of CPS, equipped with some level of computational intelligence that makes them capable of building awareness, reasoning concerning states of operations and objectives, and adapting to their environment and work conditions [28].

SCPS implement a higher level of integration of hardware, software, AI, and cyberware technologies than any other system before [29,30]. Under an I4.0 scenario, traditional CPS need to deal with the dynamicity of the environment effectively, be scalable and tolerant to threats, and control their emergent behavior; hence, CPS have to be smart. In this sense, SCPS are complex engineered systems empowered by cyber-physical computing, such as CC and FC. They tend to be smart due to AI technology, which gives them the capability of reasoning, learning, adapting, and evolving [31].

AI can provide cognition to SCPS, an essential ability to smart devices, which allows the modeling, representation, and learning of complex interactions and behaviors between the system components and the system data. Through supervised or unsupervised training, cognition can be achieved using AI models, which are designed to perform these specific tasks [32]. Moreover, AI models can continuously learn from the system, conferring an adaptive ability to the SCPS.

Within the issue of image recognition, networks have been widely used since they have demonstrated their high effectiveness in recognizing objects, detecting anomalies, and tracking objects in real-time [33]; however, there are still significant challenges in its implementation in equipment, with few hardware resources in processing and RAM primarily.

Currently, most of the intelligence of SCPS is implemented in software. For this reason, in this work, the focus was on the design of the AI software, which is one of the most complex and critical components of SCPS.

From this arises the purpose for implementing artificial neural networks in SCPS in order to integrate the learning and adaptability capabilities of neural networks that enable intelligent and autonomous decision-making.

Multilayer perceptron artificial neural network (MLPANN) was selected because of its simple and easy-to-implement architecture in comparison with other more complex neural networks [34], in addition to considering the advantages it provides in the case of application to SCPS, such as the capacity to handle large data sets, capacity to learn in a non-linear way, efficient data processing and fast training time, which makes it a viable option for real-time applications [35].

For this research, a practical experiment was conducted where MLPANN was designed and implemented on a Jupyter Notebook environment for handwritten character recognition, which can be executed and implemented to work in traditional, modern, and emerging I4.0 technologies, i.e., personal computers (PC), CC, and SCPS environments with AI embedded in FC technology to perform in IoT environments, all this in order to study and compare its effectiveness and relevance in performing this simple human task. This research experiment was developed in collaboration with the Advanced Artificial Intelligence Laboratory (LIAA), where CC and SCPS technologies were designed and implemented. The main objective of this research was to study the behavior of AI in three technologies to compare their effectiveness in embedded systems, such as SCPS, a promising emerging technology that can be applied in IoT environments to solve real problems in the I4.0 scenario.

## 2. Materials and Methods

I4.0 is a set of NTICs based, among others, on SCPS, IoT, CC, and FC technologies, which are promising for developing smart, autonomous, and intelligent systems by adding AI capabilities through ML and ANNs as shown in Figure 1.

As is shown in Figure 1, SCPS, which are becoming widespread, can be seen as an isolated application. However, Figure 2 shows that they have a synchronic integration with other cyber systems, which makes them capable of building a concise and cognitive structure that can interact deeply and autonomously with other virtual and physical systems. From this point of view, AI can play an enabling role in allowing the existence of these cognitive SCPS.

By embedding AI in CPS to make them smart, they can make decisions by themselves to optimize production schedules and processes according to the results obtained by analyzing the information obtained by sensors. This information can also feed decisions back to the production system for intelligent self-configuration and self-control; however, there are currently many challenges to obtaining these features on SCPS.

At present, most of the intelligence of SCPS is implemented in software. For this reason, in this work, the attention was focused on the AI software design, one of the most complex and critical constituents of SCP.

Consequently, this research work aimed to study and compare the performance of an MLPANN designed to solve the problem of digit recognition in three different implementation technologies: traditional (PC), modern (CC), and I4.0 emerging technology (SCPS). Comparing the performance seeks to demonstrate that AI can be embedded in CPS for developing SCPS capable of working in IoT environments.

Implementation technologies

As can be appreciated in Figure 3, three different implementation technologies were used to compare the performance of an MLPANN with an application on character recognition. The first technology was a traditional laptop personal computer (LPC). The second one was a CC environment, designed and stored on a remote server at the LIAA data center and executed through a web page on the Internet.

Finally, the last one was an SCPS, designed by LIAA, with the MLPANN embedded in fog technology and capable of working on IoT environments through mobile devices with internet connectivity through the Web.

Implementation methodology

To compare the performance of AI intelligence executed on the three proposed technologies, Figure 4 shows the designed methodology divided into four main stages: Preparing the implementation technologies, installing and configuring the operating system and software, selecting the data set, and training and testing the MLPANN.

Stage one. Implementation technologies

In this stage, to compare the performance of the ML algorithm capable of solving the digit recognition problem with MLPANNs, three different implementation technologies were chosen: LPC, CC, and SCPS.

Traditional computing (Laptop-PC): Traditional computing technology is shown in Figure 5. For the development of apps capable of working in IoT environments, its main characteristics are high performance, memory, and processing power. Its main weaknesses are the high weight, cost, size, power consumption, the lack of ports to connect sensors and actuators, and the fact that the information generated is stored locally on the hard drive. Since the equipment information can only be accessed locally, the user must be in the physical place where the equipment is located. It is also very difficult to collaborate with other machines or humans through communication networks.

Compared to CC technology, the LPC can fail, crash, or be lost; additionally, every time new software is desired, the user must install and configure it. However, fixing these problems requires highly technical knowledge, so it is often necessary to hire professionals. Currently, CC technology is replacing many of the functions performed by traditional LPC.

Modern technology (CC): The term CC is generally used to describe data centers available from anywhere to many users via the internet from any mobile or fixed device [36,37]. As shown in Figure 6, CC is the use of a network of remote servers connected to the internet to process, manage and store data, databases, servers, networks, and software [38]. Opposite to traditional LPC, instead of relying on an installed physical service, the end-user has access to a framework where software and hardware are virtually integrated.

One of the distinguishing features of CC is the on-demand availability of computer system resources, especially data storage and computing capacity, without direct active management by the user [39]. CC provides lots of processing and storage resources that can be used to analyze large amounts of data; however, the biggest drawbacks are security, reliability, and latency issues [40].

Since the end-user has no direct active management of the servers in remote locations, developing apps for working IoT environments is hard using this technology. The anterior because the end user has no access to the framework where software and hardware are virtually integrated, and because of this, it is not possible to connect sensors and actuators to the virtual machines to develop IoT applications. An alternative to solve the aforementioned problems of LPC and CC is the use of FC technology embedded in CPS, making it possible to create SCPS capable of working in IoT environments. The objective of FC is to shorten the communication paths between the cloud and the devices and reduce the data flow in external networks [41].In this work, the LIAA designed, developed, and implemented the CC remote server in its own data center, where the tests were made.

SCPS with AI embedded in FC: FC is an I4.0 emergent technology whereby data processing where part of the data emitted by the different devices connected to the network is stored in the same devices instead of being sent only to the cloud [42]. In FC, it is no longer the cloud where all the information is stored but rather a mist made up of processed data that can be both in public and/or private clouds and on the smart devices themselves. FC is closely linked to the IoT [43]. The smart devices that combine CPS, FC, and AI are capable of retaining data in such a way that they can give a faster response to the user because they do not have to connect to the network to do so. In addition, not having to access the network increases reliability for the user. The main features of SCPS with AI embedded in FC are:Lower size, weight, cost, and power consumption;Lower data consumption;Reduce internet bottlenecks, which translates into less network congestion;Consequently, lower costs and lower latency;Less bandwidth consumption;Increases the security of encrypted data since the information is closer to the end-user, reducing exposure and vulnerability;Improved scalability potential is key in a sector with so much growth potential.Decreases the reaction time of the system;The system works even when there is no internet connection, a key factor when choosing a connected system;The risk of blockages or massive failures is considerably reduced since the intelligence is not tied to a single point but is dispersed in several action nuclei;Balance and equilibrium increased;The possibility to connect sensors and actuators on Input-Output ports.

As can be seen in Figure 7, FC is a technology previous to CC where the data is generated by physical devices that have loaded the AI algorithms that are called SCPS. The FC works as several smaller decentralized data centers that will later be processed or stored in the CC.

The concept encompasses a network that extends from its boundaries, where the endpoints generate the data, to the central destination of the data in a public or private cloud site [43].

FC differs from cloud technology, above all, in the place where resources are accessed and data is processed. CC is often based on centralized data centers. Here, the servers in the backend are the ones that supply resources, such as processing power and memory, which are used by clients over the network. Communication occurs between two or more terminals through a server in the background. With concepts such as the smart factory, CC comes up against its limitations since it contains many devices constantly exchanging data. By relying on data processing close to the source, SCPS, with AI embedded in FC, manages to reduce data traffic. In this work, the LIAA designed, developed, and implemented the SCPS where the tests were made. Table 1 shows the main features of each implementation technology are hardware, price (USD), energy consumption, size, and weight.

From Table 1, it can be seen that the hardware features of the traditional LPC are the strongest; however, the main drawbacks are the cost and the energy consumption, which are the highest. It is important to highlight that the greatest weakness in implementing an IoT app with an LPC are the cost, weight, power consumption, lack of connectivity, mobility, and flexibility, high limitation characteristics for developing I4.0 technology.

The CC platform presents the greatest advantages when compared with traditional computing. Table 1 shows that the CC does not have the physical hardware equipment and that it is normal to rent a virtual server with the hardware resources and time required for the project; these will impact the cost. However, the main drawbacks are security, reliability, and latency issues and the difficulty of connecting sensors and actuators on the servers where the clouds are located. Finally, in Table 1, it can be seen that the SCPS used in this work has the weakest hardware features. However, in contrast, it has the lowest cost, power consumption, size, weight, and the possibility to integrate sensors and actuators compared to traditional and cloud computing. At the same time, SCPS integrates FC technology, a newer and disruptive version of traditional CC, which makes it possible to work on the development of technological applications in real smart IoT scenarios.

Stage two. Operating system and software

The selection of the development environment was essential to fulfill the research objective. It was necessary to install the same software in the three selected technologies to have a consistent training and testing environment for the MLPANN. For this reason, Linux was chosen as the operating system due to its customization capabilities to adapt and leverage the hardware features available. This allowed for greater compatibility in using the three technologies. It is worth noting that Linux was ideal for working in the emerging technology of SCPS, efficiently utilizing the limited memory resources, processing power, and storage available.

Python provided the necessary support for developing machine learning software and data processing. Meanwhile, Jupyter Notebook, with its browser-based execution [44], offered a consistent interface and easy access across each technology: traditional, CC, and FC.

In Figure 8, the development environment implemented in each technology is graphically depicted for training and testing the MLPANN to achieve satisfactory recognition of handwritten digits using the Modified National Institute of Standards and Technology (MNIST) handwritten digit database.

Stage three. Selecting the data set

To train and test the MLPANN on each implementation technology, the MNIST database was used, which contains a data set of 70,000 28 × 28 images of handwritten digits from 0 to 9. Pixel Format: The pixels in the images are in grayscale, meaning each pixel has a value ranging from 0 to 255, representing the grayscale intensity. The MNIST database is provided in files in a binary format. There are separate files for images and labels, where the image files contain the pixel data encoded in a binary format, and the label files contain the corresponding digit labels [45]. The MNIST is easy to use for digit recognition tasks.

Stage four. Training and testing the MLPANN

The selection of the MLPANN was mainly due to its supervised learning algorithm, a neural network architecture with a high capacity to respond to different challenges and adapt to various contexts, and its powerful nature. These characteristics make it a highly experienced network, capable of learning complex mappings and solving a wide range of machine learning problems. It is an artificial neural network feedforward that, for this experiment, was aimed at handwritten digit recognition. It was designed with three layers of nodes or neurons: 784 neurons in the input layer, 200 neurons in a hidden layer, and 10 neurons in the output layer.

Each node in the MLPANN performs a weighted sum of its inputs. The rectified linear unit (ReLU) activation function was applied in the hidden layer, and Softmax in the output layer. These functions introduce non-linearity, allowing the network to learn as the weights between nodes are adjusted to minimize the error between the predicted and desired outputs. The gradient descent optimization function was used for this purpose.

Another important parameter of this network is the learning rate, which controls the magnitude of weight adjustments during the training process. For this problem, the learning rate was set to 1 × 10^−4^. It should be noted that this value was reached through testing and adjustments in each training iteration.

For the choice of the number of epochs used in network training, the following aspects were considered:-Achieving an accuracy above 97% to ensure satisfactory recognition by the network and to be among favorable accuracy results compared to other studies on digit recognition;-Avoiding overfit and minimizing training time costs;-Achieving similar performance across the three technologies used in this research.

Several training exercises were conducted, ranging from 10 epochs to 500 epochs. Considering the aforementioned aspects, the number of epochs was adjusted in each training iteration, ultimately selecting 25 epochs as the optimal value for this experiment. The results section provides a detailed description of the disadvantages of using a higher number of epochs. This training stage of the neural network was done with 60,000 examples, and the testing one with the remaining 10,000. The model with the final selected parameters of the MLPANN can be observed in Figure 9.

After the training phase of the MLPANN, the testing phase was conducted to obtain the accuracy percentage in recognizing handwritten digits from the 10,000 images in the test set. This was calculated by evaluating the network’s performance using the mean squared error (MSE) for the three technologies.

## 3. Results

From the experiments performed, as shown in Table 2, after the training and testing stages of MLPANN in the three different technologies, it can be seen that the total percentage of MLPANN elements classified correctly in the test set is higher than 97%; the training time and precision test were also obtained:

In the first training of the MLPANN, 500 epochs were used, obtaining an accuracy of almost 98%; however, the longest training time was almost 2 h (7200 s). Analyzing these results, epoch 25 achieved a 97% digit recognition accuracy, meaning that the execution of more than 475 epochs causes a considerable increase in training time and, consequently, high energy consumption.

Considering that the MLPANN is a network that achieves a good result in a few epochs and due to the characteristics of the SCPS, such as low power consumption and processing limitations, it was decided to leave the training of the network to only 25 epochs, achieving a similar accuracy in the three technologies, as shown graphically in Figure 10, leaving the longest training time at 378 s, obtaining satisfactory results for this research.

As can be appreciated in Figure 11, compared with LPC and CC technologies, the SCPS took more time to train the network. This was mainly due to the chosen hardware used to build the SCPS for making the test; however, after the training was done, the SCPS reached the same network performance as PLC and CC. Despite a difference in the training time observed on each implementation technology, the results show that the embedded AI on the fog technology is a promising alternative for developing SCPS, which can develop smart devices for the I4.0. However, more research is needed to follow the development of this kind of I4.0 emerging technology.

Figure 12 shows some examples of successful predictions after the training and testing of the MLPANN with application in character recognition on the three implementation technologies. From Figure 12 can be seen that the MLPANN performs satisfactorily for handwritten digit recognition even in complex cases such as digits 3 and 9.

Figure 13 shows examples of predictions with false results. In the first image, it is difficult to recognize the correct result, even for a human. The second image shows the difficulty for the MLPANN to recognize the difference between 4 and 9.

In this work, the attention was focused on the AI software implementation on the SCPS to compare the performance of the MLPANN with the other implementation technologies.

The experiment is significant as its results demonstrate that current AI software development technologies can be implemented with minimal changes in emerging technologies like SCPS on FC. This brings us closer to exploring, designing, and developing SCPS in contexts with limited technical expertise and scarce hardware and economic resources while achieving important functionality such as:Data analysis, continuous learning, and adaptation to improve real-time decision-making accuracy and capacity;Real-time identification and classification of objects, patterns, and events that enable interaction with the environment and real-time adjustment to face unforeseen scenarios.

These functionalities are required in real-life scenarios, i.e., quality control systems for identifying defects or anomalies in products and triggering corrective actions without human intervention.

## 4. Discussion

Nowadays, modern society is living in the paradigm of I4.0, which is generating changes driven by NTICs, that provide high-performance computing capabilities enabling the creation of complex AI systems distinguished by being smart. Smartness is an intermittent quality of human thinking, feeling, doing, and making. Modern and emerging I4.0 engineered systems and technologies are being designed to be able to operate and provide services smartly. However, the term “Intelligent Systems” is gaining more and more traction, especially in emerging products such as networked knowledge-intensive service systems. Intelligence is becoming a necessary feature in the next generation of SCPS, which are classified according to the level of intelligence (self-awareness) and the level of organization (self-adaptation).

The SCPS integrate a certain level of computational intelligence, providing them with the capacity for reasoning and awareness regarding states of operations and objectives, as well as the ability to adapt to working conditions and environment. At present, the paradigm of SCPS is rapidly developing, making its way into the world of research and implementations, and applications are increasingly fast. Still, very few accounts exist yet on how to engineer smart systems with intelligence based on real-time event processing. A combination of SCPS, IoT, IoS FC, and ML are making I4.0 possible and the smart factory a reality.

Ultimately, it is the SCPS network integrated with the aforementioned technologies that are digitally connected and create and share information that results in the true power of I4.0. In this research, the objective was to compare the performance of the recognition of handwritten digits trained using an MLPANN in three different technologies used in the I4.0, observing each technology and its advantages and disadvantages for its work in IoT environments. After the training and testing stages of the MLPANNs, the recognition was successfully achieved with a 97% of accuracy in all cases.

The results obtained in this work show that SCPS with integrated AI on FC technology is a promising alternative for developing applications and IoT devices capable of working efficiently in I4.0 environments. However, more research is needed to develop this type of emerging technology further.

The high training time in the case of SCPS technology can be solved by using hardware with better processing and RAM characteristics. It is certainly a challenge to develop a new generation of hardware for SCPS technology with high computing resources, low power consumption, and high portability (minimal size and weight) and with advanced software that runs ML algorithms more efficiently to make a more robust ecosystem.

Traditional computing continues to be a benchmark for the design and testing of the development of ML algorithms. However, the needs of current applications, and especially those of the I4.0, demand scalable, portable environments with the use of NTICs.

Through the analysis of the three technologies, we emphasize that, indeed, the CC has a great capacity to manage processing, storage, networks, and resource management. Still, due to the large amount of data produced by IoT devices and the distance between data centers and users, the middle layer between the cloud and the IoT, FC, has become very useful.

Compared with LPC and CC technologies, SCPS have the potential of working efficiently in IoT environments, and the low weight, dimension, cost, energy consumption, portability, scalability, and physical characteristics allow connecting sensors and actuators, which are features of great relevance. Furthermore, the FC is an emerging and promising technology in which AI can be embedded for designing suitable autonomous and intelligent SCPS capable of working on more demanding I4.0 scenarios.

## 5. Conclusions

CPS may be implemented on various scales, ranging from the nano-world to large-scale systems. In this work, a CPS with the minimum characteristics was chosen to compare its performance faced with the LPC and CC technologies.

With the presented results, evidence was provided to demonstrate the advantages of SCPS compared with traditional and modern technologies; greater portability, scalability, energy savings, low cost, physical characteristics (such as small size, lightweight, low power consumption), and the possibility of adding sensors and actuators for working in IoT environments.

Furthermore, we verified that by utilizing more powerful CPS hardware for constructing SCPS, the training and testing times of ANNs can be decreased, and the performance can be enhanced for operating in more demanding Industry 4.0 scenarios.

In this research, the development of the AI algorithm was carried out using only Python. As future projects to improve the presented results in this research and expand knowledge, it is recommended to program the AI algorithms using Python’s AI development frameworks for embedded systems such as PyTorch, MicroPython, and Micro TensorFlow. This will allow for an analysis of the performance of neural network models provided by these Python tools in emerging technologies like SCPS in FC.

## Figures and Tables

**Figure 1 sensors-23-06935-f001:**
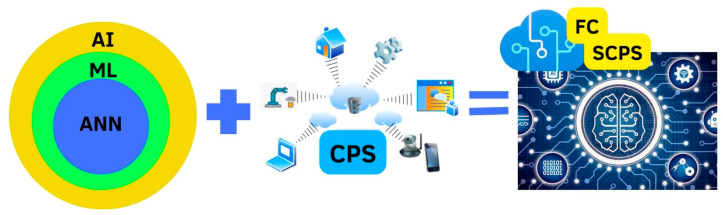
Implementation of AI embedded in CPS becomes on SCPS with added fog technology.

**Figure 2 sensors-23-06935-f002:**
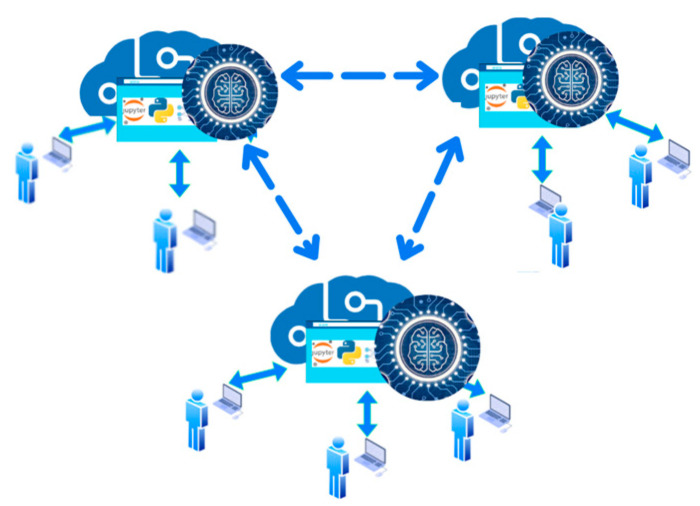
Cognitive SCPS, as a smart network, interacts in the human environment.

**Figure 3 sensors-23-06935-f003:**
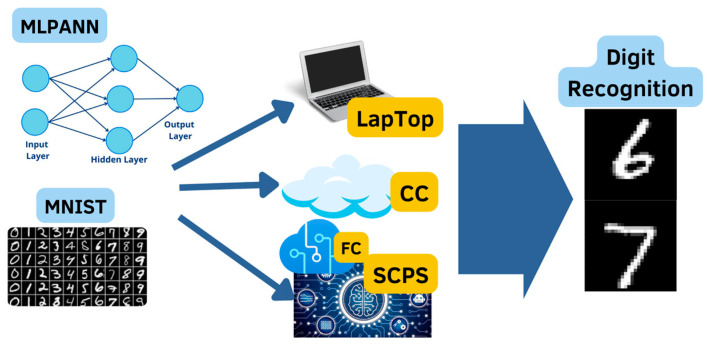
Implementation technologies to compare the performance of MLANN: LPC, CC, and SCPS.

**Figure 4 sensors-23-06935-f004:**
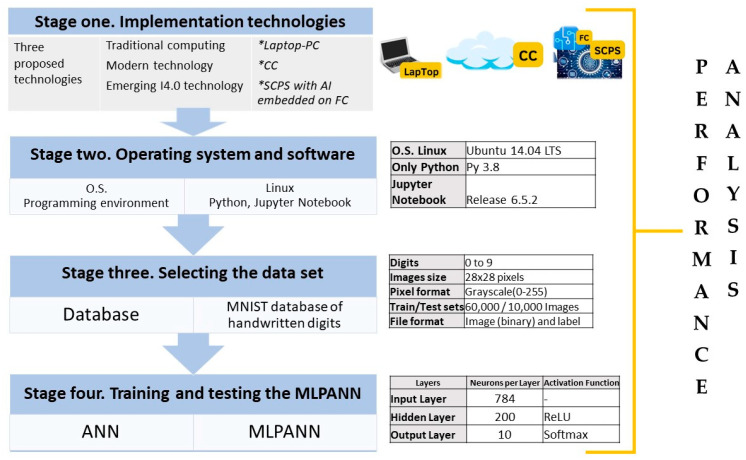
AI implementation methodology with application in digit recognition.

**Figure 5 sensors-23-06935-f005:**
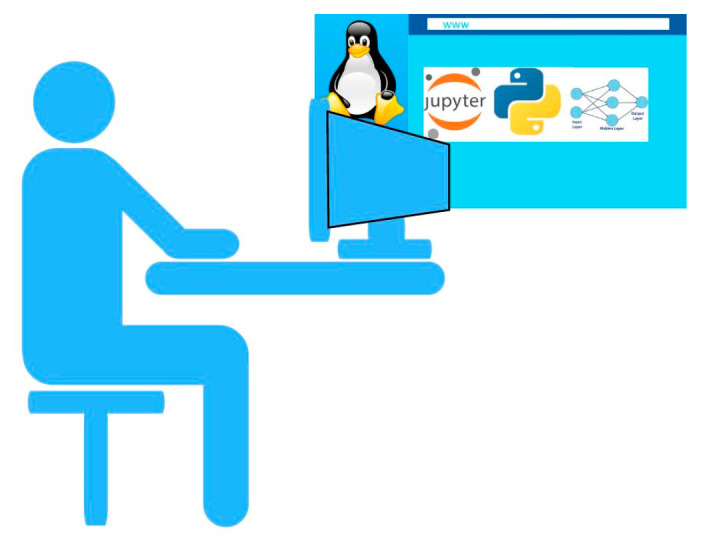
Model of traditional technology.

**Figure 6 sensors-23-06935-f006:**
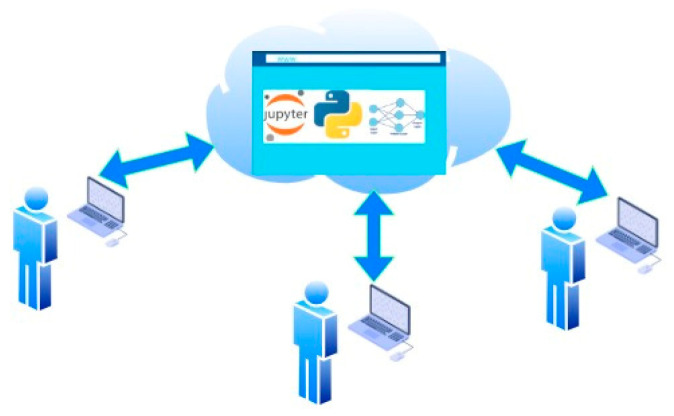
Model of CC technology.

**Figure 7 sensors-23-06935-f007:**
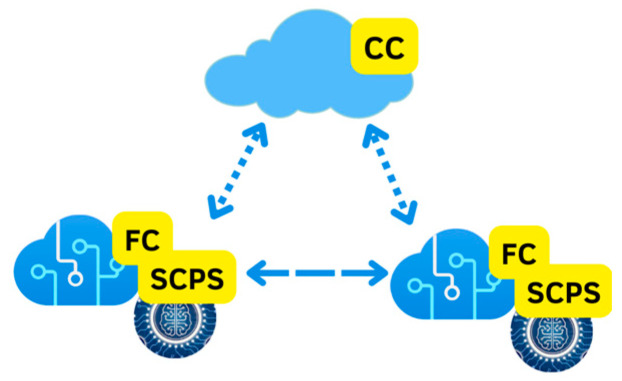
Model of fog computing SCPS technology.

**Figure 8 sensors-23-06935-f008:**
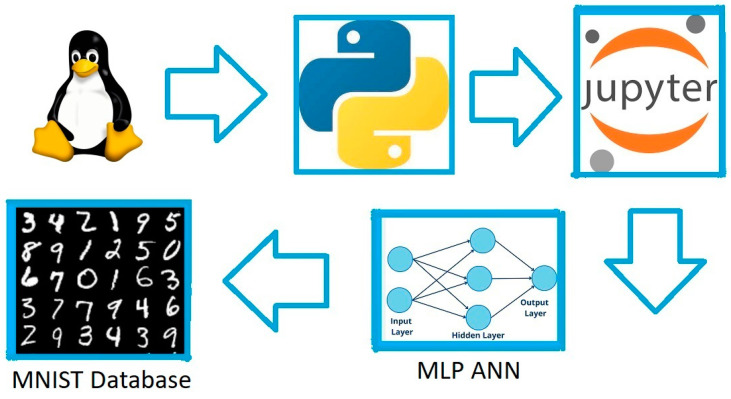
Software environment used for three I4.0 technologies.

**Figure 9 sensors-23-06935-f009:**
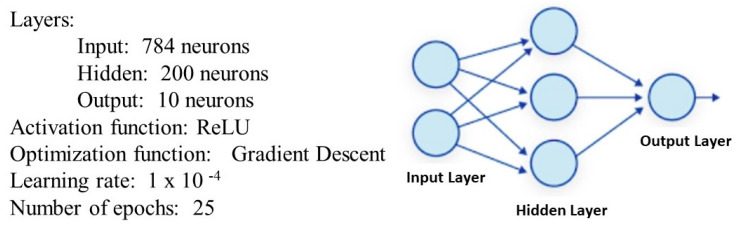
MLPANN training model.

**Figure 10 sensors-23-06935-f010:**
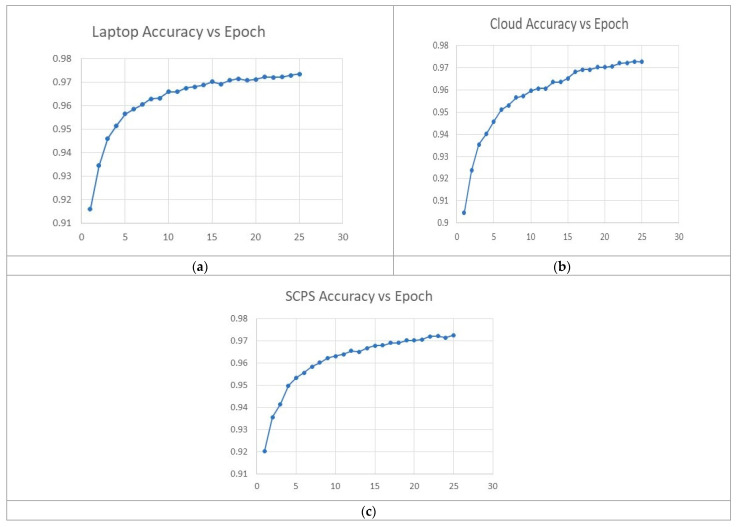
Graphs with the results of the performance of the MLPANN on: (**a**) LapTop; (**b**) cloud computing; (**c**) SCPS.

**Figure 11 sensors-23-06935-f011:**
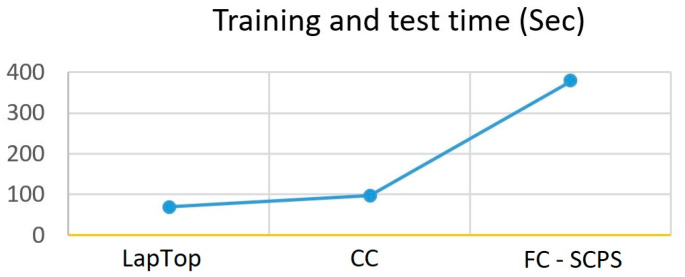
Comparison of performance in training and test time (seconds) of the three technologies.

**Figure 12 sensors-23-06935-f012:**
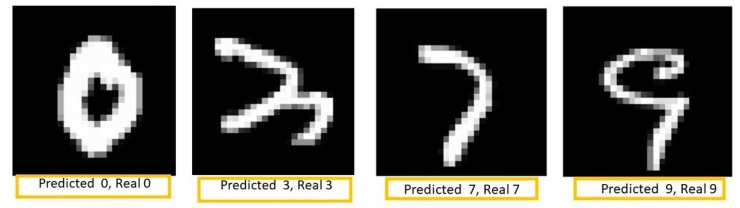
Examples of successful predictions in the test set.

**Figure 13 sensors-23-06935-f013:**
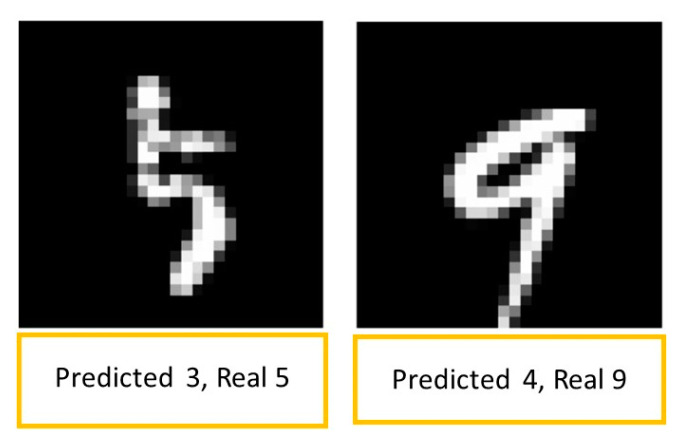
Examples of predictions with false results.

**Table 1 sensors-23-06935-t001:** Main features of the implementation technologies.

Platform	Hardware	Price (USD)	Energy Consump	Size/ Weight
Laptop	Intel Core i7-3520M processor, 2 cores, 8 GB RAM	300	65 Wh	13.31″– 9.11″– 1.34″ 4.47 lb
Cloud computing	Virtual machine with 2 cores, 4 GB of RAM	9.99 to 50 p/m	-	-
SCPS	AML-S905XCC, Minicomputer, ARM Cortex-A53 processor, 4 cores, 2 GB DDR3 of RAM	50	1.5 Wh	4.8″– 3.03″– 1.06″ 0.1124 lb

**Table 2 sensors-23-06935-t002:** MLPANN performance results of the three technologies: accuracy, training, and test time.

Technologies	Accuracy Test	Training and Test Time
Laptop	97.33%	70 s
Cloud	97.31%	97 s
SCPS	97.17%	378 s

## Data Availability

MNIST Handwritten Digit Database, Yann LeCun, Corinna Cortes and Chris Burges. Available online: http://yann.lecun.com/exdb/mnist/.

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
