# Peer review of "Performance Analysis of Embedded Multilayer Perceptron Artificial Neural Networks on Smart Cyber-Physical Systems for IoT Environments"

_sensors, 2023, doi:10.3390/s23156935_

Round 1

Reviewer 1 Report

1)    The novelty of this paper is not presented well. Please add.

2)    Introduction should provide more background on the project with the scope of the work

3)    Contribution should be clearly identified and presented under the Introduction.

4)    The paper, does not link well with recent literature on top-tier journals research gap should be clearly identified.

5)    A high-level block diagram or a framework of complete work in this paper is required to add at the beginning of chapter 2 so that readers can follow up the entire technical work that has been done in this paper. Otherwise, you can improve Figure 4 with more details about all the steps in detail from the beginning to the end of technical work.

6)    You need also to discuss the technical background of the techniques used.

7)    Results chapter is very brief. You need to expand based on the scenarios of your projects.

8)    Performance/advantages comparison with existing related works should be added at the end of the result section to validate the capability of the proposed method presented in the paper.

9)    Results section should be updated by adding a subsection as “critical analysis and discussion” where strength, limitation and impact/significance of this work in real life scenarios could be added.

10) Specific Future research directions are missing. Please add those at the end of conclusions.

Minor correction is required.

Author Response

Reply to reviewer in the attached Word file

Reviewer 2 Report

- The article discusses on an ANN approach for digit recognition deployed and tested in different setups. The existing gap in the literature and thus the contribution of the paper is not discussed, especially considering there are other approaches for this task. It is strongly suggested to condense the current introduction, stating the problem you are trying to solve, the existing gap in the state of the art, and your contribution.

- In addition, it is not clear what practical problem this study is trying to solve. Is it aimed as a theoretical study or it targets a real-world problem? Please clarify.

- The literature review is focused on cyber physical systems, while one would expect to review different methods for character recognition and their comparison. As a result, it is not clear why ANNs were selected in this study and assess the arguments on why they were the right choice. IT is suggested to re-write it, with a clear link to the main contribution of this paper, clearly presenting the gaps and summarizing on this study's contribution.

- it seems only the MNIST dataset was used for training and testing the ANN, but without testing any additional handwritten or otherwise created characters, thus testing the proposed approach in a real-world situation.

The word digitization seems to be used out of context. Please check replacing with digitalization.

Author Response

(The authors gave the same response as above.)

Reviewer 3 Report

"Performance Analysis of Embedded Multilayer Perceptron Artificial Neural Networks on Smart Cyber-Physical Systems for IoT Environments" is a well-executed study that explores the performance of MLP networks in IoT settings.

While the paper presents promising experimental results, it would have been beneficial to include a real-world deployment validation of the proposed MLP network approach. This would provide insights into the challenges and limitations faced when implementing the network on actual smart cyber-physical systems.

The paper briefly mentions the MLP network architecture used in the experiments but lacks a detailed discussion on the rationale behind the specific architecture selection. Elaborating on the considerations for choosing the architecture, such as the number of layers and nodes, would enhance the paper's completeness.

The paper could benefit from expanding the discussion on the implications of the findings and their practical applications. Providing practical insights and recommendations for integrating MLP networks into existing IoT frameworks would enhance the relevance and impact of the research.

Author Response

(The authors gave the same response as above.)

Round 2

Reviewer 1 Report

I would like to thank the authors for addressing my comments. I have no other comments.

Reviewer 2 Report

No further comments

Reviewer 3 Report

-